# Inflammatory Responses during Tumour Initiation: From Zebrafish Transgenic Models of Cancer to Evidence from Mouse and Man

**DOI:** 10.3390/cells9041018

**Published:** 2020-04-20

**Authors:** Abigail Elliot, Henna Myllymäki, Yi Feng

**Affiliations:** UoE Centre for Inflammation Research, Queen’s Medical Research Institute, University of Edinburgh, 47 Little France Crescent, Edinburgh EH16 4TJ, UK; a.elliot-1@sms.ed.ac.uk (A.E.); Henna.Myllymaki@ed.ac.uk (H.M.)

**Keywords:** zebrafish, inflammation, tumour initiation, macrophage, neutrophil, cancer, pre-neoplastic, live imaging, tumourigenesis, tumour model

## Abstract

The zebrafish is now an important model organism for cancer biology studies and provides unique and complementary opportunities in comparison to the mammalian equivalent. The translucency of zebrafish has allowed in vivo live imaging studies of tumour initiation and progression at the cellular level, providing novel insights into our understanding of cancer. Here we summarise the available transgenic zebrafish tumour models and discuss what we have gleaned from them with respect to cancer inflammation. In particular, we focus on the host inflammatory response towards transformed cells during the pre-neoplastic stage of tumour development. We discuss features of tumour-associated macrophages and neutrophils in mammalian models and present evidence that supports the idea that these inflammatory cells promote early stage tumour development and progression. Direct live imaging of tumour initiation in zebrafish models has shown that the intrinsic inflammation induced by pre-neoplastic cells is tumour promoting. Signals mediating leukocyte recruitment to pre-neoplastic cells in zebrafish correspond to the signals that mediate leukocyte recruitment in mammalian tumours. The activation state of macrophages and neutrophils recruited to pre-neoplastic cells in zebrafish appears to be heterogenous, as seen in mammalian models, which provides an opportunity to study the plasticity of innate immune cells during tumour initiation. Although several potential mechanisms are described that might mediate the trophic function of innate immune cells during tumour initiation in zebrafish, there are several unknowns that are yet to be resolved. Rapid advancement of genetic tools and imaging technologies for zebrafish will facilitate research into the mechanisms that modulate leukocyte function during tumour initiation and identify targets for cancer prevention.

## 1. Introduction

It has been well documented by pathologists that tumour masses are often densely packed with cells of both adaptive and innate arms of the immune system. In 1986, Dvorak first drew the comparison between tumour formation and wound healing, famously describing the tumours as ‘wounds that do not heal’ [1]. More recently, Hanahan and Weinberg have highlighted the significance of inflammation as an enabling hallmark in their updated version of ‘Hallmarks of Cancer’ [2]. Inflammatory cells within the tumour microenvironment play a role in promoting tumour progression and metastasis and a high index of innate immune cell infiltration is often associated with poor prognosis [3,4]. Moreover, it is known that inflammation caused by pre-existing chronic inflammatory conditions conveys a predisposition to cancer development (the ‘extrinsic pathway’), whilst genetic events leading to neoplasia themselves promote the recruitment of inflammatory cells into tumours (the ‘intrinsic pathway’) [5]. However, it is only in recent years that in vivo live imaging studies have captured the initial inflammatory response to tumour initiation; revealing the rapid recruitment of macrophages and neutrophils in response to the oncogenic transformation of pre-neoplastic cells (PNCs) [6]. More importantly, these studies show that this intrinsic inflammation begins to exert a trophic influence on PNC growth even at this nascent stage of cancer development [6,7]. These findings demonstrate the advantages of using zebrafish to model cancer and the power of live imaging approaches to uncover novel aspects of cancer biology. In particular, zebrafish models facilitate the study of cellular dynamics during the initial phase of tumour development, a stage previously thought intractable to study.

The zebrafish has been a major vertebrate model for developmental biology since the 1960s. Several forward genetic screens lead to the identification of mutants affecting almost every organ and cell type, and most of the causative genes identified were found to be conserved in the human [8]. The complete zebrafish reference genome shows that 70% of the human genome has at least one conserved orthologue in zebrafish [9]. Naturally occurring neoplastic lesions have been observed in wild type zebrafish reared under laboratory conditions [10], and major cancer-related genes that affect humans are conserved in zebrafish, allowing for their direct comparison, e.g. the master tumour suppressor, p53 [11], and components of the MAPK pathways [12]. Due to their high level of conservation, in addition to their fecundity, ease of genetic manipulation, and translucency at the larval stage, the zebrafish is now an important player in cancer research.

In this review we discuss the benefits of transgenic zebrafish models that facilitate in vivo live imaging studies of tumour initiation. We describe the studies conducted in zebrafish that have allowed the field to establish the significance of early host inflammatory responses in promoting cancer development at the pre-neoplastic stage, with a focus on innate myeloid cells. We highlight the signals required for the recruitment of macrophages and neutrophils to PNCs and the heterogeneous nature of their responses. Furthermore, we summarise evidence from the literature that suggests these inflammatory components are conserved in early mammalian tumourigenesis. To do so, we make comparisons with studies of inflammation within early mouse neoplastic lesions where available, and also draw upon studies of tumour-associated macrophages (TAMs) and tumour-associated neutrophils (TANs) in the later stages of mammalian cancer. Whilst tumour infiltrating lymphocytes are also key components of the tumour microenvironment in mammalian cancer [13,14,15], there is little evidence that lymphocytes play a role during tumour initiation. The role of lymphocytes during pre-neoplastic development is also yet to be explored in zebrafish cancer models, largely because mature lymphocyte subsets have only recently been characterized in zebrafish. Finally, we discuss interesting avenues for future research and bring attention to recent technical advances within the zebrafish field that will facilitate further research of the pro-tumour inflammatory response and the potential discovery of cancer prevention strategies.

## 2. Zebrafish Transgenic Models of Cancer for In Vivo Live Imaging Studies of Tumour Initiation

Zebrafish larvae develop most major organs, vasculature and a fully functional innate immune system within the first 5 days post fertilization. Most significantly, they are naturally transparent. This transparency, combined with fluorescent labelling, allows real-time observation of single cells in a live in vivo model. Furthermore, by deleting genes required for pigmentation, the *Casper* strain has been created, which remains translucent throughout adulthood [16]. The most evident benefit of using zebrafish as a cancer model is the capacity for in vivo live imaging. For example, the first transgenic zebrafish model for cancer was a T-cell leukaemia model, which was established by the expression of the mouse homologue of oncogene c-myc, tagged with fluorescent GFP, under the control of the T-cell specific promoter, Rag2 [17]. This model allowed direct monitoring of the initiation and expansion of leukaemic cells from the thymus by fluorescence microscopy [18].

Since the establishment of the Tol2 transposase transgenesis protocol for zebrafish, generation of transgenic strains has become a routine procedure [19,20]. This has accelerated the development of zebrafish cancer models, many of which mirror human disease in terms of both histopathological features and molecular signatures. For example, zebrafish melanoma models expressing the common melanoma oncogenes BRAF^V600E^, NRAS^Q61K^ and HRAS^G12V^ under the melanocyte-specific *mitfa* promoter, or HRAS^G12V^ under the *kita* promoter, are all sufficient to drive melanoma tumour formation, either alone or in combination with p53^−/−^ mutation [21,22,23,24]. These models all recapitulate human melanoma with respect to their hyperpigmentation, histology and where tested, their transcriptomic gene expression profiles. The expression of human oncogenes under the control of tissue specific promoters has been employed to create representative cancer models for various organs, including the skin [23], intestine [25], pancreas [26] and brain [27] (See Table 1).

Zebrafish models of cancer commonly feature fluorescently tagged oncogenes, which label cancer cells and allow real time monitoring of tumour promotion and progression. Furthermore, the creation of ‘transgenic reporters’ also allows the visualisation of gene expression, i.e. by using the promoter region of a gene of interest to drive the expression of a fluorescent protein. As such, zebrafish models offer a unique contribution to the field of cancer biology, by capturing events which cannot be directly observed in mammalian in vivo models, such as tumour initiation and metastasis. This review focuses upon the use of zebrafish for the study of tumour initiation, while the application of zebrafish for the study of metastasis has been reviewed by Osmani and Goetz, 2019 [28].

Tumour initiation is the first stage of cancer development, during which normal cells undergo oncogenic transformation, i.e. genetic changes that enable them to form tumours. The combination of live imaging with genetic and chemical manipulations in zebrafish has allowed the dissection of novel mechanisms involved in tumour initiation. For example, transgenic reporters have been utilised to study the genetic mutations and key signalling pathways that contribute to tumour initiation in pancreatic cancer and neuroblastoma [29,30]. A central question with respect to tumour initiation is why only a small proportion of clonal pre-neoplastic cells survive and go on to form tumours. Multiple zebrafish models have been used to explore the fate of pre-neoplastic cells, uncovering a mechanism by which de-differentiation conveys a tumour initiating fate. By combining live imaging with a transgenic reporter for the developmental gene, crestin, the de-differentiation of single clones within pre-neoplastic lesions was observed, followed by an acceleration of tumourigenic activity leading to melanoma formation [31]. A related phenomenon was also observed in a zebrafish model of pancreatic tumour initiation, in which labelling tumour-initiating cells with a marker of differentiation revealed that oncogene expression blocked differentiation, leading to tumourigenesis [26]. Interestingly, live imaging has also led to the observation that cells expressing oncogenic Ras or v-Src can be extruded from the epidermis by healthy neighbouring cells, indicative of a potential tumour-suppressive defence mechanism [32,33,34].

The main limitation in the use of transgenic models to study tumour initiation is that oncogene expression is dependent upon tissue-specific promoters, for which timing of activation can vary. This also restricts the study of early events to larval zebrafish. However, inducible systems for transgene expression have been appropriated from the *Drosophila* and mouse fields, including the Tet/On system [35], the Lex/PR system [36], the tamoxifen-inducible GAL4/UAS system [37] and the heat-shock-inducible Cre/Lox system [38]. These systems have recently been used to develop inducible cancer models, which now enable temporal precision for the study of tumour initiation in both larval and adult fish (see Table 1) [25,39,40,41].

In addition to the use of zebrafish for the study of cancer biology, the zebrafish has also been widely used for the study of haematopoiesis [42] and the innate immune response [43,44,45]. Both macrophages and neutrophils share comparable developmental origins with their mammalian counterparts [46,47,48,49], and exhibit a high degree of functional conservation, for example, with respect to host–pathogen interactions [50] and wound healing [51,52]. Considering the importance of inflammation in the development and progression of cancer, zebrafish researchers have turned their attention to the study of macrophages and neutrophils in relation to cancer. Fluorescent transgenic reporter lines for both macrophages and neutrophils have been developed using macrophage-specific promoters, mpeg1.1 [53] or mfap4 [54], and neutrophils-specific promoters, mpo [55] or lyz [56], (see Table 2). In this way, the dynamics of the immune response can be directly imaged, allowing live in vivo visualisation of the interactions between leukocytes and cancer. Studies combining zebrafish cancer models with leukocyte markers have demonstrated that zebrafish macrophages and neutrophils have tumour-promoting roles comparable to their mammalian counterparts. Furthermore, modelling tumour initiation in zebrafish has elucidated mechanisms of tumour-promoting inflammation which had never before been captured at this early stage.

## 3. Macrophages and Neutrophils are Co-Opted by Cancer to Perform Tumour-Supporting Activities

High levels of macrophages and neutrophils in cancerous lesions correlate with poor prognosis in humans. The pro-tumour activities of these leukocytes are evident in both mammalian and zebrafish models.

### 3.1. Tumour-Associated Macrophages

Tumour-associated macrophages (TAMs) are the most abundant type of leukocyte found within tumours, comprising up to 50% of the tumour mass in epithelial tumours [82,83]. The degree of macrophage infiltration correlates with poor clinical prognosis across a broad range of tumour types [3]. Both tissue-resident macrophages and blood monocyte-derived macrophages infiltrate tumours, where they are influenced by signals derived from cancer cells and the local tumour microenvironment to perform tumour-promoting activities [84]. The activation state of TAMs more closely resembles that of ‘alternative’ or ‘anti-inflammatory’ M2 polarization, as opposed to ‘classical’ or ‘pro-inflammatory’ M1 polarization, e.g. genes commonly expressed by TAMs include scavenger receptors, arginase-1, matrix metalloproteinases, TGF-β and IL-10, accompanied by downregulation of pro-inflammatory cytokines and MHC Class II [85,86]. Although, it is of note that the concept of M1/M2 polarisation is an oversimplification and TAMs are a highly heterogenous population [3,87,88,89]. The tumour-supporting roles of TAMs have been studied extensively in the past two decades, revealing mechanisms by which TAMs promote proliferation, angiogenesis, invasion and metastasis [84,90,91,92]. In zebrafish cancer xenograft models, zebrafish macrophages have also been shown to promote angiogenesis, invasion and metastasis [93,94].

### 3.2. Tumour-Associated Neutrophils

Neutrophils are frequently found within tumours and, following the convention of TAMs, the term ‘tumour-associated neutrophil’ (TAN) has been adopted. However, TANs are less well characterised in comparison to their TAM counterparts and have been shown to have both pro- and anti-tumour effects [95]. Nonetheless, in numerous studies a high intra-tumour neutrophil density has been associated with poor patient prognosis [96,97]. Multiple tumour-promoting functions of TANs have been described, including the encouragement of proliferation, angiogenesis, metastasis and immunosuppression [95,98,99]. In addition, through the secretion of genotoxic nitric oxide and reactive oxygen species (ROS), neutrophils exert a mutagenic effect, and thus provide a further driver for cancer development. On the other hand, higher levels of nitric oxide release from neutrophils can be cytotoxic for cancer cells, contributing to tumour suppression [100,101]. Further anti-tumour roles of neutrophils may include triggering apoptosis via activation of TRAIL [102] and promoting the activity of anti-tumour cytotoxic T lymphocytes [103]. In zebrafish, tumours also exhibit high levels of infiltrating neutrophils, which promote metastasis and express genes involved in angiogenesis and immunosuppression [104,105,106,107]. Anti-tumour neutrophil responses are yet to be described within zebrafish cancer models.

In addition to mature neutrophils, under pathogenic conditions such as chronic inflammatory disease and cancer, immature myeloid cells, known as myeloid-derived suppressor cells (MDSCs), are recruited from the bone marrow prior to terminal differentiation [85,108]. Granulocytic MDSCs (G-MDSCs) are thought to be immature neutrophils since they morphologically and phenotypically resemble neutrophils. G-MDSCs promote tumour cell survival, angiogenesis, invasion, metastasis, and immunosuppression; functions similar to those of mature pro-tumour TANs [109]. In contrast, existing studies have not described anti-tumour functions for G-MDSCs [110]. It is of note that not all cancer studies make a clear distinction between mature TANs and immature G-MDSCs, and there is a lack of clarity as to whether they are independent cell types or whether they originate from the same progenitors [110,111,112]. The existence of MDSCs has not been explored in zebrafish and it is uncertain whether neutrophils in zebrafish cancer studies are recruited in a mature or immature state.

## 4. Evidence of Tumour-Promoting Inflammation in Early Tumourigenesis of Mouse and Man

Whilst there is abundant evidence that TAMs and TANs have tumour-promoting roles within established tumours, relatively little is known about the role of macrophages and neutrophils at the earliest stage of tumour initiation. This is largely because the early stages of tumourigenesis are difficult to detect in humans and mammalian models. The exception to this is colorectal cancer, for which distinctive structures known as ‘adenomatous polyps’ are formed at the early neoplastic stage [113]. These polyps are easy to detect and their removal is a routine procedure. Furthermore, the common initiating mutation, APC, was discovered over 30 years ago by studying families with familial adenomatous polyposis (FAP). FAP is caused by an inherited mutation in the APC gene, leading to abundant polyp formation and, without intervention, malignant disease [114,115]. This discovery was swiftly followed by the creation of APC mutant mouse models which represent not only FAP but also spontaneous colorectal cancers, 80% of which feature mutations in APC [116,117]. The ability to detect and model early neoplastic legions in this way enabled the discovery of factors which promote early tumourigenesis, such as prostaglandin E_2_ (PGE_2_)

PGE_2_ is a potent inflammatory mediator, synthesised by the COX-2 enzyme. Both PGE_2_ and COX-2 are present at high levels in colorectal cancer, adenomatous polyps and even pre-neoplastic microadenoma [118,119]. Chemical inhibition of COX-2, as well as the deletion of the COX-2 gene or various PGE_2_ receptors, was found to dramatically decrease intestinal polyp formation in numerous mouse models of colorectal cancer, including APC mutant mice and mice exposed to carcinogens [120,121,122,123,124]. Non-sterile anti-inflammatory drugs (NSAIDs), such as sulindac and COX-2 inhibitor celecoxib, have been tested for the prevention of both FAP and spontaneous colorectal cancers and have proven to effectively reduce polyp formation and prevent cancer [125,126,127]. Epidemiological studies have also shown that the use of COX-2 inhibitor, aspirin, is associated with decreased incidence of colorectal cancer [128]. Unfortunately, the use of COX-2 inhibitors to prevent cancer is limited by the dangerous cardiovascular side effects that are associated with long-term use [129].

PGE_2_ is also implicated in the promotion of many other types of cancer, including cancer of the breast [130], liver [131], lung [132], brain [133] and pancreas [134,135]. This suggests that the phenomenon by which inflammation fuels early tumour promotion may be a common feature of cancer development. Therefore, the discovery of alternative methods to target this early inflammatory response is imperative for cancer prevention. Inhibiting the recruitment of either macrophages or neutrophils also dramatically reduced polyp formation mouse models of colon cancer and prevented carcinogenesis [136,137,138,139]. This demonstrates the importance of inflammatory leukocytes in promoting early tumourigenesis. As such, both macrophages and neutrophils are promising targets for cancer prevention, but further research is required to understand precisely when and how along the pathway of oncogenesis leukocytes are recruited and co-opted by the cancer cell.

Research in this field has been impeded by a lack of suitable in vivo models. Most mouse models for the study of tumourigenesis rely on read-outs such as tumour size and occurrence, whilst measurements of the early pre-neoplastic stage are largely missing. These models also lack temporal resolution and the means to dissect complex mechanisms involving interactions between multiple cell types. For these reasons, the zebrafish is a uniquely advantageous model for the study of inflammation during early tumourigenesis.

## 5. Zebrafish Studies Reveal Leukocyte Recruitment and Trophic Function at the Pre-Neoplastic Stage

Studies utilising transparent zebrafish larvae to model tumour initiation were the first to report a trophic inflammatory response at the pre-neoplastic stage. Feng et al. 2010 first showed that leukocytes are rapidly recruited following the expression of a single oncogene [6]. Subsequent zebrafish studies have shown a similarly rapid recruitment of neutrophils and/or macrophages to the skin, liver and brain in response to transgenic expression of numerous oncogenes, e.g. HRAS^G12V^, KRAS^G12V^, NRAS^Q61K^, Src, Xmrk, Myc and Akt [6,7,39,40,41,60,68,104,140,141,142,143]. Live imaging has provided insight into the dynamics of the intrinsic inflammatory response following tumour initiation within these models. Furthermore, the depletion of leukocytes has revealed that both macrophages and neutrophils have a potent tumour-promoting effect as early as the pre-neoplastic stage. Here we highlight the details of these findings within each tissue type:

### 5.1. Skin

In vivo live imaging with fluorescent leukocyte markers in the *Tg(kita:Gal4TA, UAS:eGFP-HRAS^G12V^)* melanoma model showed that leukocytes are recruited to PNCs at the earliest stage of tumour initiation, even prior to the expansion of single clones [6]. Retention of leukocytes in the PNC microenvironment was sustained over time, indicative of chronic non-resolving inflammation. The up-regulation of pro-inflammatory cytokines in PNC-bearing larvae further confirmed an early onset of inflammation in response to tumour initiation. Hydrogen peroxide (H_2_O_2_), a potent chemoattractant at wounds, was identified as the local chemoattractant guiding neutrophils towards PNCs. This discovery was facilitated by the use of live imaging, which allowed a direct comparison of neutrophil dynamics between tumour initiation and wounding, and the use of a fluorescent H_2_O_2_ probe to visualise secretion from PNCs. Although both neutrophils and macrophages were observed to engulf material from PNCs, pan-leukocyte depletion (by Gcsfr + PU.1 morpholino) lead to a greatly decreased number of PNCs demonstrating their pro-tumour function [6,7]. Neutrophil depletion alone (using Gcsfr morpholino) lead to greater reduction of PNCs than macrophage depletion alone (using Irf8 morpholino), suggesting that neutrophils have a greater contribution to trophic inflammation in this model. Furthermore, PGE_2_ was found to be one of the trophic factors released by leukocytes, directly promoting PNC proliferation through the EP1 receptor [7]. It is of note, however, that exogenous PGE_2_ only partially rescued PNC numbers in pan-leukocyte-depleted larvae, which suggests that leukocytes produce additional, non-redundant trophic signals.

A more precise temporal resolution of the inflammatory response to tumour initiation was observed by tamoxifen-inducible transformation of keratinocytes with GFP-tagged HRAS^G12V^. Here, neutrophils were recruited within 8 “hours post induction” (hpi) and leukocyte depletion resulted in a 50% reduction of PNC number at 48 hpi [144]. By using correlative light and electron microscopy (CLEM) to study leukocyte recruitment in this model, it was observed that the dominant route by which immune cells gain access to PNCs is through proteolysis-independent breaching of weak spots within the basement membrane. However, proteolysis-dependent damage to the basement membrane was seen underneath areas of PNC clonal expansion, which suggests that PNCs can have invasive properties. Indeed, HRAS^G12V^-transformed keratinocytes expressed the invasion-related genes Mmp9 and Slug [104]. PNCs also expressed the chemokine, IL-8, which was shown to attract neutrophils via Cxcr2. Interestingly, reduced neutrophil recruitment also partially attenuated the expression of Mmp9 and Slug in PNCs, suggesting that neutrophils encourage the invasive behaviour of PNCs. Transcriptomic analysis of PNC-associated neutrophils in an NRAS^Q61K^ melanoma model, showed upregulation of genes that correspond with growth- and invasion- promoting properties, e.g. FGF-1 and -6, cathepsin-H and galectin1 -and -3 [60].

### 5.2. Liver

Using the Tet-On system to induce KRAS^G12V^ expression in the hepatocytes of zebrafish larvae triggered a trophic inflammatory response in the liver [41]. As seen in response to tumour initiation in the skin, neutrophils were recruited at approximately 8 hpi. Upon entering the liver, neutrophil motility was decreased, indicative of a chronic inflammatory response. Such retention behaviour has also been observed for TANs in mouse tumour models [145]. Following the induction of KRAS^G12V^, both PNCs and neutrophils expressed high levels of IL-1β and TGF-β. Abrogation of TGF-β signalling reduced liver size, decreased the number of infiltrating neutrophils, and triggered the upregulation of pro-inflammatory genes (IL-1β, TNFα, IL-6, IL-8, IL-12), indicating that TGF-β is a key modulator of neutrophil function in the pre-neoplastic microenvironment of the liver.

The Tet-On inducible system also allowed Yan et al. (2017) to explore the role of both neutrophils and macrophages in response to tumour initiation in the adult fish liver [140]. PNCs expressed both the neutrophil chemoattractant, IL-8, and the macrophage chemoattractant, CSF-1, and an array of pro-inflammatory genes were upregulated in both macrophages and neutrophils (IL-1β, Cxcl1b, Nfkb2, CSF-1, IL-6 and IL-8). Leukocyte depletion completely attenuated the increase in PNC proliferation and survival observed in leukocyte-bearing larvae, restoring the size of the liver to that of wild-type. Depletion of macrophages had a comparable effect compared to the depletion of neutrophils; both resulted in a partial attenuation of liver enlargement and equivalent reduction in PNC proliferation, suggesting that both cell types provide important trophic signals. A cortisol-dependent sex bias in disease severity was observed in male fish consistent with that observed in human hepatocellular carcinoma (HCC). TGF-β expression and the levels of both macrophages and neutrophils in the liver were strongly correlated with this bias, further implicating TGF-β as a mediator of pro-tumour inflammation.

Transformation with other common HCC oncogenes, Xmrk and Myc, also elevated the expression of TGF-β in PNCs and triggered neutrophil and macrophage recruitment to the liver [141]. This demonstrates that oncogene-dependent inflammation is not specific to KRAS^G12V^. β-catenin overexpression also caused oncogenic transformation in hepatocytes, with associated recruitment of macrophages and neutrophils [78]. Whilst the independent influence of oncogene-dependent inflammation is yet to be determined in this model, a combination of β-catenin expression and high fat diet induced a strong tumour-promoting inflammatory response, which was associated with TNFα positive macrophages. Finally, modelling a rare type of liver cancer (fibromellar carcinoma), by transgenic expression of a DnaJ-PKAc fusion protein, resulted in leukocyte recruitment to the liver [146]. Here too, an increase of TNFα positive macrophages was observed, alongside a liver-wide activation of caspase-1, indicating inflammasome activation and IL-1β release. Chemical inhibition of either TNFα or caspase-1 reduced neutrophil and macrophage numbers to wild-type levels, demonstrating the potency of these cytokines as recruitment factors.

### 5.3. Brain

Glioblastoma can be modelled in the zebrafish by transgenic overexpression of Akt in neural cells, a gene commonly upregulated in human glioblastoma [142]. Oncogenic Akt transformation lead to an overabundance of microglia in the brain. These microglia exhibited reduced mobility, sustained contact with PNCs and a distinctive amoeboid morphology indicative of activation. Live imaging combining both macrophage- and microglia-specific reporters showed that the increase in microglia numbers was due to a recruitment of peripheral macrophages, which subsequently differentiated into microglia upon entry into the brain parenchyma. Treatment with anti-inflammatory drug, dexamethasone, or CSF-1R inhibitor reduced macrophage recruitment and PNC proliferation. Complete abrogation of macrophage recruitment by Irf8^−/−^ or Cxcr4^−/−^ resulted in even greater suppression of PNC proliferation. This demonstrates that macrophages/microglia have a potent trophic role during glioblastoma initiation and Cxcr4 is required for their recruitment.

Interestingly, microglia-mediated trophic support to PNCs in the brain might be cell-contact dependent. By live imaging with a transgenic Ca^2+^ reporter, Chia et al. (2019) discovered that PNC-microglia interactions are governed by Ca^2+^-dependent ATP release from PNCs and detection of extracellular ATP by the P2y12 receptor on microglia [143]. This mechanism is normally employed under physiological conditions that govern recruitment of microglia to highly active neurons [143,147,148,149,150]. Reduction of Ca^2+^ levels, inhibition of ATP release, or inhibition of the P2y12 receptor all significantly abrogated PNC-microglia interactions and greatly reduced the rate of PNC proliferation, equivalent to that of macrophage/microglia depletion described above [142,143]. Therefore, ATP-P2y12 signalling, and possibly direct PNC-microglia contact, were required for the tumour-promoting effect of Akt-dependent inflammation in the brain.

Whilst Chia et al. (2018) reported an absence of neutrophils in the brain during Akt-dependent tumour initiation, in contrast, neutrophil recruitment into the brain was observed by Powell et al. (2018) following KRAS^G12V^ transformation of astrocytes [68,142]. Although there is a discrepancy between these studies, both peripheral macrophage and neutrophils are found within human brain tumours [151]. In this case, KRAS^G12V^-expressing astrocytes also expressed high levels of IL-8, and blocking neutrophil recruitment by abrogation of Rac2 or Cxcr1 significantly reduced the proliferation of pre-neoplastic astrocytes [68]. This suggests that neutrophils have a trophic influence upon tumour initiation in the brain, and that IL-8-Cxcr1 signalling is required for neutrophil recruitment.

## 6. Mechanisms Governing Intrinsic Pro-Tumour Inflammation: Parallels between Zebrafish, Mouse and Man

Overall, the above zebrafish studies have identified numerous signals involved in the activation and recruitment of inflammatory leukocytes in response to tumour initiation (see Figure 1), in addition to trophic signals that feedback on PNC proliferation. Although equivalent studies of the pre-neoplastic stage do not exist in mammalian models, there is evidence that these signals are present at early stages of mammalian tumourigenesis (see Table 3). Mammalian studies also provide mechanistic clues as to how these signals govern tumour-promoting inflammatory responses.

### 6.1. Pro-Inflammatory Cytokines are Expressed in Response to Tumour Initiation

Zebrafish skin and liver cancer models showed an upregulation of the pro-inflammatory cytokines IL-1β and TNFα in response to tumour initiation [6,7,68,78,140,146]. Both of these inflammatory cytokines were important for the recruitment of leukocytes to the liver [146].

In mouse models of skin and colon cancer knock-out of IL-1β/IL-1R or downstream signal transducer, MyD88, lead to a dramatic and sustained abrogation of leukocyte recruitment, in addition to a reduced occurrence of colon polyps and skin papillomas [152,153,154]. This demonstrates that IL-1β is required for instigating tumour-promoting inflammation during early tumourigenesis in mice. IL-1β activates and amplifies host inflammatory responses by activation and recruitment of innate immune cells via expression of pro-inflammatory cytokines and chemokines [155,156]. In support of this mechanism in response to tumour initiation, in vitro studies have shown that IL-1β, TNFα and IL-6 expression are induced by transformation with oncogenic forms of Ras in various human and mouse cell types [154,157,158,159,160,161].

Knock-out of TNFα or TNFR1 also reduced leukocyte infiltration and increased resistance to tumour development in mouse models of skin and liver cancer [162,163,164]. TNFα signalling is well-established to upregulate pro-inflammatory cytokines and chemokines, and can itself act as a chemoattractant signal, which may explain the requirement of TNFα for leukocyte recruitment in both zebrafish and mouse studies [165]. Dichotomously, TNFα signalling can also directly induce apoptosis of cancer cells, or promote cell survival through activation of NF-κB signalling [165]. Although the direct effect of TNFα upon PNCs has not been explored in zebrafish models of tumour initiation, the latter mechanism is more likely since no signs of leukocyte-mediated cell death have been recorded and TNFα positive macrophages promote tumourigenesis in zebrafish liver cancer models [140,146].

### 6.2. Chemokines Recruit Leukocytes to PNCs

Zebrafish models of both brain and liver cancer have indicated that CSF-1 contributes to the recruitment of macrophages in response to tumour initiation [140,142]. Likewise, CSF-1 signalling is a well-established macrophage recruitment signal in mammalian cancers [166,167,168,169,170]. With respect to early tumourigenesis in mice, abrogation of CSF-1 signalling reduced tumour formation in mouse models of thyroid cancer and pancreatic neuroendocrine cancer [171,172]. However, blocking CSF-1 signalling in breast cancer models only affected the progression to invasive metastatic carcinoma [173]. Similarly, in the development of mouse colon cancer, lack of CSF-1 lead to a modest reduction in the number of small polyps but a dramatic reduction in the number of large polyps, suggesting a role in promoting later progression [174]. Therefore, the involvement of CSF-1 during tumour initiation might be tissue-specific. In addition to its role in macrophage recruitment, CSF-1 contributes to the control of the M2 phenotype of mammalian TAMs; both pre-clinical and clinical studies have shown that inhibition of CSF-1 signalling does not always deplete TAMs from established tumours, but rather impairs their tumour-promoting functions [170,175]. This suggests that other factors also drive the recruitment and retention of macrophages in cancer.

Chia et al. (2018) showed that, whilst CSF-1 played a role in macrophage recruitment, signalling to Cxcr4 was absolutely required for the recruitment of macrophages to the Akt-transformed brain in zebrafish tumour initiation, and that PNC-derived Cxcl12 was the major ligand for this interaction [142]. Cxcr4 signalling has been shown to promote progression of many cancers in mammals including glioblastoma, through both cancer cell autonomous signalling and leukocyte interactions [106,176,177,178,179,180,181]. However, the pro-tumour effect of Cxcr4 signalling at the preneoplastic stage was macrophage specific [142]. In the mouse APC^min^ model of colorectal cancer, Cxcr4^−/−^ mice exhibited reduced macrophage infiltration and polyp formation [182]. Furthermore, in the mouse Lewis lung carcinoma model, inhibition of Cxcr4 greatly reduced tumour-associated inflammation and tumour growth [183]. Thus, there is some evidence that Cxcr4 signalling recruits macrophages and has a pro-tumour effect during early stage tumourigenesis in mammalian models. Cxcr4 has also been implicated in the recruitment of neutrophils and G-MDSCs to cancer cells in mouse xenograft models, resulting in tumour-promoting effects [182,184,185,186].

IL-8 was expressed in zebrafish PNCs in response to oncogenic Ras expression [140,187] and IL-8 signalling through Cxcr1/2 chemokine receptors was shown to drive pro-tumour neutrophil recruitment at the pre-neoplastic stage in zebrafish models of skin and liver cancer, contributing to PNC proliferation and sustained tumour-promoting inflammation [68,104]. Likewise, Ras-induced expression of IL-8 promotes neutrophil recruitment and tumourigenesis in mouse xenograft models [159,188,189]. IL-8 has also been found to provoke a neutrophil-dependent anti-tumour effect in mouse models, consistent with the ability of TANs to perform both pro- and anti- tumour functions [190]. More broadly, Cxcr1/2 signalling has been shown to recruit pro-tumour TANs and/or G-MDSCs in a variety of mammalian cancers [191,192,193,194,195]. With respect to early tumourigenesis, Cxcl1 and Cxcl2 have been shown to drive neutrophil and G-MDSC infiltration in mouse models of colorectal cancer, promoting polyp and tumour formation [137,196,197]. These findings suggest that signalling via the IL-8 - Cxcr1/2 chemokine axis promotes tumour initiation.

### 6.3. Plasticity and Heterogeneity of Leukocytes

The expression of pro-inflammatory cytokines by macrophages and neutrophils in response to tumour initiation in zebrafish studies corresponds with the hypothesis that tumour-associated inflammation begins with a pro-inflammatory response, which becomes progressively more anti-inflammatory throughout the course of tumourigenesis [86,87,198,199,200,201]. This phenomenon mimics the inflammatory wound response; in both zebrafish and mammals the wound response is characterised by early M1/N1 activation followed by a switch to M2/N2 activation states, the latter of which promotes tissue repair and resolution [51,202,203,204]. The exception here is that tumours provoke a sustained response, thus creating a positive-feedback cycle of non-resolving inflammation and tissue repair [198]. Interestingly, macrophages and neutrophils within zebrafish studies had trophic effects at the pre-neoplastic stage (as early as 24 h following oncogenic transformation), and in the presence of pro-inflammatory signals [6,7]. This suggests that, in the case of intrinsic inflammation, the M2/N2 switch occurs at the pre-neoplastic stage. The simultaneity of pro-inflammatory and trophic signals may be achieved by an intermediate phenotype between M1/N1 and M2/N2, and/or heterogeneity of leukocyte responses. In support of the latter, leukocyte heterogeneity was observed in zebrafish studies of tumour initiation; only 20% of leukocytes recruited to PNCs in the skin expressed PGE synthase, whilst minor populations of macrophages were positive for either M1 (TNFα) or M2 (arginase-1) markers [6,7]. Furthermore, in response to oncogenic transformation in the liver, only 35% of macrophages were TNFα positive [146].

### 6.4. PGE_2_ as Trophic Factor and Immunomodulator

As previously discussed, PGE_2_ is a central component of pro-tumour inflammation in humans and mice, and has been identified as a potent leukocyte-derived trophic signal at the pre-neoplastic stage in zebrafish [7]. In support of this finding, PGE_2_ is also secreted by macrophages within neoplastic polyps in humans and mice [205,206,207]. PGE_2_ signalling has a direct trophic function in mouse models of colon cancer, for example, through activation of β-catenin [128]. Macrophage-derived PGE_2_ also increases COX-2 expression in neoplastic cells, creating a positive feedback cycle, which greatly increases the local level of PGE_2_ in the colon, driving tumourigenesis [208].

Within mammalian immune responses and cancer, COX-2 expression is upregulated by both pro-inflammatory cytokines, such as IL-1β, and tumour promoters, such as Ras [209,210,211,212]. PGE_2_ itself is a central inflammatory mediator governing both positive and negative regulation of inflammation through alternative receptors [213,214]. Whilst PGE_2_ enhances early inflammation, it later exerts an immunosuppressive effect on both macrophages and neutrophils [215,216,217,218,219]. Thus, PGE_2_ may also contribute to shaping the inflammatory response at the pre-neoplastic stage. For example, the immunosuppressive effect of PGE_2_ could prevent neutrophils from performing anti-tumour activities and may contribute to the M2/N2 phenotype of pro-tumour leukocytes. Indeed, COX-2 inhibitors have been shown to repolarize M2 TAMs [220,221,222], and PGE_2_ has been implicated in the induction of MDSCs in tumour-bearing mice [223,224,225].

### 6.5. TGF-β Governs Pro-Tumour Neutrophils

In zebrafish models of liver tumour initiation, high levels of TGF-β expression by PNCs are associated with both disease severity and the expression of anti-inflammatory genes by pro-tumour neutrophils [41,141]. The mechanism has also been indicated in mammals by treatment of mouse tumours with a TGFBR inhibitor, which resulted in a reduction of neutrophils and reduced tumour size [226]. These neutrophils showed an upregulation of pro-inflammatory cytokines, chemokines and iNOS, and had a direct cytotoxic effect on tumour cells due to ROS secretion. Although TGF-β signalling can itself affect cancer cell survival, the reduction in tumour size was neutrophil- dependent. Further studies in mouse models of cancer have also implicated TGF-β as a regulator of pro-tumour G-MDSCs. These studies showed that anti-TGFβ antibodies had a therapeutic effect by depleting MDSCs in a breast cancer model [227], and myeloid-specific deletion of TGFBR2 reduced tumour metastasis in breast, lung and melanoma models [228]. More recently, the role of TGF-β in the recruitment of pro-tumour neutrophils has been described in mouse models of hepatocellular carcinoma through Cxcl5, the expression of which corresponds to disease severity in human patients [229]. Whilst these studies do not represent tumour initiation, they do indicate that TGF-β has a pro-tumour effect in mammalian cancer, including liver cancer, via recruitment and modulation of neutrophils.

### 6.6. Extracellular ATP Attracts Pro-Tumour Microglia/Macrophages

Chia et al. (2019) showed that the trophic effect of microglia during glioblastoma tumour initiation was dependent upon Ca^2+^-mediated ATP release from PNCs and signalling through the P2y12 receptor [143]. This represents a repurposing of an endogenous mechanism found in both zebrafish and mice, which governs microglia recruitment to areas of high calcium, such as injury and seizures [147,148,149,150]. Similarly, in a mouse model of glioma, extracellular ATP signalling through the purinergic receptor, P2X7R, promotes the recruitment of macrophages and microglia [230]. Although microglia are quite divergent in comparison to other macrophage types [231], extracellular ATP signalling though purinergic receptors also modulates the function of macrophages in inflammatory responses outside of the brain and promotes M2 polarisation in TAMs [232,233,234]. Therefore, this represents a mechanism for further study with regards to tumour initiation in other tissue types.

## 7. Future Perspectives

Zebrafish cancer models have shown that PNC-induced inflammation is an important player with respect to tumour initiation. Hitherto, the role of inflammatory leukocytes at the pre-neoplastic stage was largely unknown. Imaging of transparent zebrafish larvae has demonstrated that neutrophils and macrophages are recruited in response to tumour initiation and both have a trophic effect upon PNCs. The ever-expanding collection of transgenic reporters, combined with efficient methods for genetic manipulation in the zebrafish, provide a toolbox for the future analysis of cellular and signalling responses within the developing PNC niche. As such, the zebrafish model will provide much needed insight to support the development of novel strategies for cancer prevention.

Thus far, zebrafish studies have identified some of the chemoattractants, inflammatory mediators and trophic factors that contribute to PNC-induced inflammation. Whilst these signalling molecules correspond with pro-tumour responses in mouse and man, further research is required to confirm their mechanisms of action at the pre-neoplastic stage. There also remain additional signals that are yet to be discovered, for example, it is apparent that PGE_2_ is not the sole leukocyte-derived factor influencing PNC proliferation at this stage [7]. Furthermore, the mechanisms that govern the pro-tumour phenotype of macrophages and neutrophils remain unclear. A starting point to tackle this question would be to study known factors involved in polarising TAMs and TANs in the tumour microenvironment for a role at the PNC niche. Chemical inhibitors for blocking TAM recruitment and/or M2 polarisation are currently in clinical trials for cancer treatment and may also be promising for cancer prevention [235]. Considering the pro-tumour role of neutrophils in zebrafish studies of tumour initiation, targeting neutrophils may also be an effective approach. Moreover, a more potent effect may be achieved by developing strategies to reprogram macrophages and neutrophils to an ‘M1’ or ‘N1’ state, wherein they may partake in anti-tumour responses. Some promise has been shown in this area, regarding the reprogramming of TAMs in mouse models [175,220,235,236], but this is yet to be explored in zebrafish. Traditional methods for studying leukocyte polarisation in mammalian immunology involve the analysis of multiple cell surface markers by flow cytometry, a method which is not compatible with zebrafish due to a lack of zebrafish-specific antibodies. However, the recent development of single-cell RNA-sequencing technologies provides a powerful solution. This approach will provide insight into macrophage and neutrophil heterogeneity within the PNC microenvironment and identify marker genes that can be used for the development of novel transgenic reporters. It will also drive the discovery of novel factors that mediate trophic interactions between leukocytes and PNCs.

In addition to promoting PNC proliferation, zebrafish cancer models have also provided evidence that genes involved in angiogenesis and invasion are expressed at the pre-neoplastic stage by both PNCs and inflammatory cells [60,104,140,144]. This is of interest since these processes are traditionally associated with later stages of cancer but are likewise difficult to capture at early stages in mammalian models [2]. This presents angiogenesis and invasion, and their relationship with inflammation, as subjects for further study in zebrafish tumour initiation models. Dissecting complex multicellular mechanisms such as these will be facilitated by the ease of genetic manipulation in the zebrafish. For example, a system for cell-type specific CRISPR-mediated gene depletion has recently been developed [237,238]. This will help to pick apart the functions of specific cell types and the roles of pleiotropic genes.

Alongside innate inflammatory cells, tumour-infiltrating lymphocytes also play key roles in the tumour microenvironment [13,14,15]. Furthermore, recent advances in immunotherapy have revolutionized cancer treatment by promoting the adaptive immune response against cancer cells [239,240]. However, little is known as to when and how adaptive immune cells respond to pre-neoplastic or early neoplastic stage tumour development. Zebrafish may prove a useful model for addressing this question. The developmental origin, anatomical location and genetics underlying the development of adaptive immunity are largely conserved between zebrafish and mammals [241,242,243,244,245,246]. In recent years, studies of zebrafish lymphocytes have also revealed the existence of functionally conserved mature T cell subsets, such as CD4^+^ helper T cells [247,248], T regulatory cells [247,249,250] and γδ T cells [251]. The characterisation of mature lymphocyte subpopulations in zebrafish remains an ongoing topic of research, with an expanding collection of fluorescent reporter lines, e.g. pan T cell reporter, *Tg(lck:eGFP)* [244]; CD4^+^ T cell reporter, *Tg(CD4-1:mCherry)* [247]; and regulatory T cell reporter, *Tg(foxp3a:eGFP*) [249,250]. This will allow researchers to explore the role of T cells during early tumour development using in vivo live imaging.

Other common components of the tumour microenvironment include extracellular vesicles (EVs) and stromal cells, which may also play a role at the pre-neoplastic stage. EVs are known to transmit signals that modulate the behaviour of recipient cells. Cancer-derived EVs have been shown to promote M2 polarisation of TAMs in some cases, and the expression of pro-inflammatory cytokines in others [252,253,254]. Recently, both a fluorescent probe [255,256] and a transgenic line [257] have been developed that specifically label EVs in zebrafish, making it possible to track their transit in a live in vivo model. For example, one study showed that tumour-derived EVs activated macrophages, resulting in a macrophage-dependent promotion of metastatic outgrowth at distal sites [255]. These tools will be invaluable in addressing the question of whether PNC-derived EVs modulate host inflammatory cell function. Stromal cells within the PNC microenvironment are also of interest. ‘Cancer-associated fibroblasts’ have been identified as cancer-promoting agents in mammalian tumours [258], but their role at the pre-neoplastic stage remains largely unexplored. One study has recently shown that leukocytes recruited in response to tumour initiation in the liver release serotonin, which activates mesenchymal stromal cells and encourages them to secrete TGF-β, amplifying pro-tumour inflammation [259]. This demonstrates a tumour-promoting role for mesenchymal stromal cells during liver tumour initiation and suggests a direction for future study in other tissues.

As well as detailed mechanistic studies, the zebrafish model is also amenable to high-throughput screening. The small size of zebrafish embryos allows imaging of the whole organism and several individuals can be screened simultaneously. For example, Precazzini et al. (2019) developed an automated, imaging-based screen with 384-well plates to detect modifiers of melanocyte hyper-proliferation in a zebrafish melanoma model [260]. Taking advantage of existing fluorescently tagged cancer models, it would be possible to perform in vivo imaging-based screening for cancer preventative chemicals. The development of highly efficient methods for CRISPR-mediated gene deletion in first generation larvae, known as ‘crispants’, also provides the means to carry out high-throughput CRISPR screening for the discovery of novel targets [238]. Thus, the zebrafish is a powerful model with the potential to reveal mechanisms governing pre-neoplastic lesion development, and to test strategies for cancer prevention.

## Figures and Tables

**Figure 1 cells-09-01018-f001:**
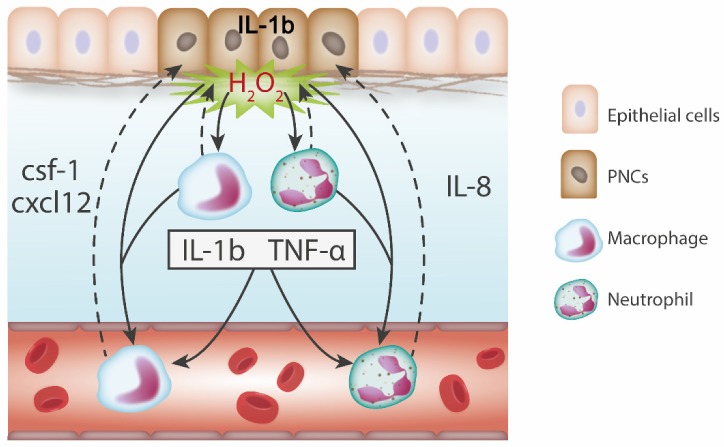
Signals governing the recruitment of innate immune cells in response to tumour initiation. IL-1β is secreted by PNCs in response to oncogenic transformation, activating local immune cells, which in turn express inflammatory cytokines such as IL-1β and TNFα. H_2_O_2_ released from PNCs acts as a local chemoattractant for the recruitment of inflammatory leukocytes. CSF-1 and Cxcl12 contribute to the recruitment of macrophages through interaction with CSF-1R and Cxcr4 respectively, these signals are both PNC and macrophage derived. IL-8, expressed by both PNCs and neutrophils in response to tumour initiation, is responsible for neutrophil recruitment via Cxcr1/2.

**Table 1 cells-09-01018-t001:** Established Transgenic Zebrafish Cancer Models. The following models have been shown to develop tumours with pathological features similar to their mammalian counterparts.

Organ	Cell Type	Promoter	Oncogene	Marker	Regulation	Ref.
**Blood**	Lymphoblasts	Xef1a	(*Hs*) ETV6-Runx1	eGFP	Promoter-driven	[57]
		Actb2	(*Hs*) ETV6-Runx1	eGFP	Promoter-driven	[57]
	T-Lymphoblasts	Rag2	(*Mm*) c-Myc	eGFP	Promoter Driven	[17]
**Skin**	Melanocytes	Mitfa	(*Hs*) BRAF^V600E^, p53^−/−^	None *	Promoter-driven	[21]

			(*Hs*) BRAF^V600E^, p53^−/−^	eGFP	Promoter-driven	[58,59]
			(*Hs*) HRAS^G12V^	GFP	Promoter-driven	[23]
			(*Hs*) HRAS^G12V^	mCherry	Promoter-driven	[23]
			(*Hs*) NRAS^Q61K^	mCherry	Inducible LexPR	[60]
	Melanocytes and Goblet Cells	KITa	(*Hs*) HRAS^G12V^	eGFP	GALTA4/UAS	[24]
			(*Hs*) NRAS^Q61K^	mCherry	Inducible LexPR	[60]
**Liver**	Hepatocytes	Fabp10	(*Xl*) pt-β-cat	None *	Promoter-driven	[61]
			(*Dr*) KRAS^G12V^	eGFP	Promoter-driven	[62]
			(*Dr*) KRAS^G12V^	eGFP	Inducible LexPR	[63]
			(*Dr*) KRAS^G12V^	eGFP	Inducible Tet-On	[64]
			(*X*) Xmrk	None *	Inducible Tet-On	[65,66]
			(*Mm*) c-Myc	None *	Inducible Tet-On	[66,67]
**Intestine**	-	Fabp2	(*Dr*) KRAS^G12V^	eGFP	Inducible LexPR	[25]
**Brain**	Glial cells	Gfap	(*Hs*) KRAS^G12V^	mCherry	GAL4/UAS	[27]
			(*Hs*) KRAS^G12V^	GFP	Promoter-driven	[68]
**Brain & PNS**	Glial cells	Krt5	(*Hs*) KRAS^G12V^	mCherry	GAL4/UAS	[27]
**Adrenal Gland**	Neuroblasts	Dβh	(*Hs*) n-Myc	eGFP	Promoter-driven	[30]
**Pancreas**	Progenitor cells	Ptf1a	(*Hs*) KRAS^G12V^	eGFP	Promoter-driven	[26]
			(*Hs*) KRAS^G12D^	eGFP	GAL4/UAS	[29]
**Pituitary Gland**	Corticotrophs	Pomc	(*Dr*) PTTG	None *	Promoter-driven	[69]
**Muscle**	Progenitor cells	Rag2	(*Hs*) KRAS^G12D^	None *	Promoter-driven	[70,71,72]
		Cdh15	(*Hs*) KRAS^G12D^	None *	Promoter-driven	[72]
		Mylz2	(*Hs*) KRAS^G12D^	None *	Promoter-driven	[72]

* Fluorescent marker can be incorporated by crossing with suitable reporter line, e.g. *Tg(mitfa:eGFP)*. PNS = peripheral nerve sheath. Promoters: Krt5 (Cytokeratin 5); Gfap (Glial fibrillary acidic protein); Fabp10 (Fatty acid-binding protein 10); Rag2 (Recombination activating gene 2); Xef1a (Xenopus laevis elongation factor 1a); Actb2 (Zebrafish β-actin 2); Mitfa (Melanocyte inducing transcription factor a); KITa (KIT proto-oncogene receptor tyrosine kinase a); Dβh (Dopamine-β-hydroxylase); Ptf1a (Pancreas Associated Transcription Factor 1a); Cdh15 (Cadherin 15); Mylz2 (Myosin light chain, phosphorylatable, fast skeletal muscle 2). Species: *Hs* (*Homo sapiens*); *Mm* (*Mus musculus*); *Xl* (*Xenopus laevis*); *Dr* (*Danio rerio*); *X (Xiphophorous)*. Oncogenes: KRAS (Kirsten rat sarcoma viral oncogene homolog); pt-β-cat (β-catenin S33A, S37A, T41A and S45A); Myc (myelocytoma proto-oncogne); ETV6-Runx1 (fusion of ETS variant transcription factor 6 and runt-related transcription factor 1); BRAF (proto-oncogene, serine/threonine kinase B-Raf); HRAS (Harvey rat sarcoma viral oncogene homolog); NRAS (neuroblastoma RAS viral oncogene homolog); PTTG (pituitary tumour transforming gene).

**Table 2 cells-09-01018-t002:** Zebrafish Transgenic Reporter Lines for Innate Immune Cells. The following transgenic lines are used to label innate immune cells by the studies referenced within this review, this is by no means an exhaustive list of available reporter lines.

	Promoter	Marker	Notes	Ref.
**Neutrophil**	Mpx/Mpo	GFP	-	[73]
		eGFP	-	[55]
		mCherry	-	[74]
		eGFP-L10a	Ribosomes and polysomes	[75]
		BirA-Citrine	Biotin-tagging	[76]
	LysC/Lyz	dsRed	-	[77]
		eGFP	-	[56,77]
		BFP	-	[78]
**Macrophage**	Mpeg1.1	eGFP	-	[53]
		mCherry	-	[53]
		mCherry-F	Membrane Bound	[79]
		Dendra	Photoconvertible	[80]
		CFP-DEVD-YFP	FRET, caspase cleavable	[81]
		BirA-Citrine	Biotin-tagging	[60]
	Mfap4	tdTomato-CAAX	Membrane Bound	[54]
		Turquoise2	-	[54]
		dLanYFP-CAAX	Membrane Bound	[54]

Promoters: Mpx (Myeloperoxidase); LysC (Lysozyme C); Mpeg1.1 (Macrophage expressed gene 1.1); Mfap4 (Microfibril Associated Protein 4).

**Table 3 cells-09-01018-t003:** Signals governing the trophic inflammatory response to tumour initiation in zebrafish are conserved in mouse models of early neoplastic development.

	Zebrafish (Pre-neoplastic Stage)	Mouse (Early Neoplastic Stage)
H_2_O_2_	Released by PNCs and neighbouring cells in the skin [6].	Unknown
	Promotes PNC proliferation and local leukocyte recruitment [6].	
IL-1β	Expressed in response to tumour initiation in the skin, liver and brain [6,41,68,140,146].	Promotes neoplasm formation and leukocyte recruitment in the skin and colon [152,153,154].
	Promotes PNC proliferation and leukocyte recruitment in the liver [146].	Upregulates pro-inflammatory cytokines and growth factors [155].
TNF-α	Expressed in response to tumour initiation in the skin and liver [6,78,140,146].	Promotes neoplasm formation and leukocyte recruitment in the skin and liver [162,163,164].
	Promotes PNC proliferation and leukocyte recruitment in the liver [146].	
CSF-1	Expressed in response to tumour initiation in the liver [140].	Promotes neoplasm formation and macrophage recruitment in thyroid and pancreas [171,172].
	Promotes PNC proliferation and leukocyte recruitment in the brain [142].	Only promotes later stages of breast and colon cancers [173,174].
Cxcl12-Cxcr4	Cxcl12 is expressed in response to tumour initiation in the brain [142].	Promotes neoplasm formation and macrophage recruitment in colon and lung [182,183].
	Promotes PNC proliferation by macrophage recruitment [142].	
IL-8- Cxcl1/Cxcl2	IL-8 is expressed in response to tumour initiation in the skin, liver and brain [6,41,68,104,140].	Promotes neoplasm formation and the recruitment of neutrophils and G-MDSCs in the colon [137,196,197].
	Cxcr1/2 signalling promotes PNC proliferation and neutrophil recruitment in the brain and skin respectively [68,104].	
PGE_2_	Produced by leukocytes in response to tumour initiation in the skin [7].	Secreted by neoplastic cells and macrophages in the colon [205,206,207,208].
	Directly promotes PNC proliferation [7].	Directly promotes proliferation [128].
TGF-β	Expressed in response to tumour initiation in the liver [41,104,140,141].	Unknown
	Governs neutrophil phenotype and promotes PNC proliferation [41].	
ATP	Released by PNCs in response to tumour initiation in the brain [143].	Unknown
	Promotes PNC proliferation and microglia contact via purinergic signalling [143].	

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
