# Peer review of "Inflammatory Responses during Tumour Initiation: From Zebrafish Transgenic Models of Cancer to Evidence from Mouse and Man"

_cells, 2020, doi:10.3390/cells9041018_

Round 1
Reviewer 1 Report
The authors review the trophic roles of innate immune cells on tumour initiation in zebrafish and mouse models. It’s well written and worth publishing in Cells. But I am appreciate it if you could prepare a table showing which discoveries belong to zebrafish or / common in mouse studies, especially in Section 6. It will be a great help to make readers understand your review easily.
Minor issues
Table 1.
About “mitfa:BRAFV600E/p53-/-” fish (Marker: N/A), Richard White’s group created the same transgenic lines with several types of fluorescent markers. Please refer their efforts.
I know it’s a minute thing, but please distinguish “eGFP” and “GFP”.
Figure 1.
Please refer about Figure 1 in the main text.
In Section 7, the font size of some sentences is different. Please correct.
Reference
Please correct #127 info.
#226. MemBright probe seems good for zebrafish research, but the reference did not state about it. I don’t think it’s appropriate reference.
Reviewer 2 Report
The authors Elliot A. et al. submitted a review focused on the advantages of the zebrafish model in the study of inflammation in tumor initiation processes. They describe the features of the adult and embryo zebrafish models and compare those with common mammalian models. Moreover, it is reported the implication of different immune system cell populations in the tumorigenesis processes and the conservation of these mechanisms also in zebrafish. The authors closed the manuscript describing the future implications of this innovative in vivo model in the field of tumor inflammation.
The review is quite well written and useful for scientists involved in this field of research. For all these reasons the review could be suitable for publication after major revisions
Major concerns
1- The authors describe the characteristics of many cell populations belong to the immune system in zebrafish and the relative transgenic strains for each component. The manuscript does not describe the development of the immune system during embryogenesis, from the early stages to the adult fish. In the first part of the review, the authors should report the origin of these cell populations during the embryogenesis.
2- In the manuscript is well described the role of TAM and TAN during the tumor initiation processes. The text lacks to report the role of lymphocytes in the tumor initiation processes and in zebrafish. The interest of the scientific community on lymphocytes and cancer has been increased in recent years due to the impact of the immunotherapy on patient’s treatment. The study of lymphocytes activities is complicated in the embryo model, but the authors should report the features of these cell populations in zebrafish and their implications in the tumor inflammation.
Round 2
Reviewer 2 Report
Now the manuscript is acceptable for pubblication on "Cells"